# Multilevel analysis of predictors of multiple indicators of childhood vaccination in Nigeria

**Justice Moses K. Aheto** [1,2]*, **Oliver Pannell**[1], **Winfred Dotse-Gborgbortsi**[1], **Mary K. Trimner**[3], **Andrew J. Tatem**[1], **Dale A. Rhoda** [3], **Felicity T. Cutts**[4], **C. Edson Utazi**[1]

**1** WorldPop, School of Geography and Environmental Science, University of Southampton, Southampton, United Kingdom, **2** Department of Biostatistics, School of Public Health, College of Health Sciences, University of Ghana, Accra, Ghana, **3** Biostat Global Consulting, Worthington, OH, United States of America, **4** Department of Infectious Disease Epidemiology, London School of Hygiene and Tropical Medicine, London, United Kingdom

\* j.m.k.aheto@soton.ac.uk, justiceaheto@yahoo.com

**Data Availability Statement:** DHS data are publicly available from https://dhsprogram.com/data/available-datasets.cfm. Other data (i.e., geospatial covariates) are publicly available via the sources

## Abstract

### Background

Substantial inequalities exist in childhood vaccination coverage levels. To increase vaccine uptake, factors that predict vaccination coverage in children should be identified and addressed.

### Methods

Using data from the 2018 Nigeria Demographic and Health Survey and geospatial data sets, we fitted Bayesian multilevel binomial and multinomial logistic regression models to analyse independent predictors of three vaccination outcomes: receipt of the first dose of Pentavalent vaccine (containing diphtheria-tetanus-pertussis, *Hemophilus influenzae* type B and Hepatitis B vaccines) (PENTA1) (n = 6059) and receipt of the third dose having received the first (PENTA3/1) (n = 3937) in children aged 12–23 months, and receipt of measles vaccine (MV) (n = 11839) among children aged 12–35 months.

### Results

Factors associated with vaccination were broadly similar for documented versus recall evidence of vaccination. Based on any evidence of vaccination, we found that health card/document ownership, receipt of vitamin A and maternal educational level were significantly associated with each outcome. Although the coverage of each vaccine dose was higher in urban than rural areas, urban residence was not significant in multivariable analyses that included travel time. Indicators relating to socio-economic status, as well as ethnic group, skilled birth attendance, lower travel time to the nearest health facility and problems seeking health care were significantly associated with both PENTA1 and MV. Maternal religion was related to PENTA1 and PENTA3/1 and maternal age related to MV and PENTA3/1; other significant variables were associated with one outcome each. Substantial residual community level variances in different strata were observed in the fitted models for each outcome.

referenced in the methods section, and also presented in the Supplementary Information Table S2. The authors did not have any special access privileges that others would not have.

**Funding:** This work was supported by funding from the Bill and Melinda Gates Foundation (Investment ID INV-003287). CEU and AJT received the grant. The funder did not play any role in the study design, data collection, analysis and interpretation of data, the report writing, and the decision to submit the manuscript for publication.

**Competing interests:** The authors have declared that no competing interests exist.

## Conclusion

Our analysis has highlighted socio-demographic and health care access factors that affect not only beginning but completing the vaccination series in Nigeria. Other factors not measured by the DHS such as health service quality and community attitudes should also be investigated and addressed to tackle inequities in coverage.

## Introduction

Childhood vaccination is one of the core strategies for achieving goal 3 of the Sustainable Development Goals (SDGs) of reducing under-five mortality to less than 25/1000 live births by 2030 [1]. Immunization Agenda 2030 (IA2030) [2] has the goal of all persons fully benefiting from vaccines to improve overall health and general well-being. IA2030 preconizes extension of immunization services to regularly reach "zero-dose" (those who receive no vaccines–often approximated by those who did not receive the first dose of pentavalent vaccine (containing diphtheria-tetanus-pertussis, *Hemophilus influenzae* type B and Hepatitis B vaccines)–PENTA1) and under-immunized children and to advance and sustain high coverage for all vaccines across the life course.

The global challenge of getting vaccines to all children is highlighted in the case of Nigeria. Notably, in 2019 Nigeria had the highest estimated number of infants who did not receive PENTA1 and the highest number of those who did not receive MV through routine services [3].

To increase vaccine uptake and implement optimal intervention strategies to address childhood vaccination inequalities in Nigeria, factors (both enablers and barriers) that predict vaccination coverage in children should be identified and addressed in a timely manner. Several studies, especially those conducted in sub-Saharan Africa have identified different individual, household, and community level factors to be associated with childhood vaccination. These include maternal education and age, household wealth, mother's attendance for antenatal care, skilled birth attendance, rural/urban residence, knowledge of immunization, shorter distance from vaccination site, level of partner's support, trust in vaccines and immunization programs, lifestyle, number of children, occupation, lack of time or language barriers, and reminders to parent's [4–15].

In this study, we develop both binomial and multinomial multilevel statistical frameworks to identify factors that predict select indicators of childhood vaccination in Nigeria to help policymakers and program managers make informed decisions aimed at improving the survival and health of children [12,13,16,17]. In particular, we investigate potential differences in factors that predict vaccination according to the strength of evidence of vaccination (documented versus recall evidence of vaccination), and examine whether factors affecting whether children start the vaccination series differ from those related to completing it, having begun.

We examine the role of geospatial environmental and climatic (geospatial) factors as well as individual, household and additional community level attributes in predicting vaccination coverage using multilevel modelling approaches which acknowledge the nested structure of the data, i.e., child, household, community and stratum levels. Although geospatial covariates have been extensively used for predicting vaccination coverage and other health outcomes in children [18–20], there is insufficient evidence on how geospatial factors combine to influence vaccination at the individual child level, especially when adjusting for other covariate effects [19,21,22]. This is one of the objectives we aim to address in this work.

## Materials and methods

### Data

The study used data from the nationally representative cross-sectional 2018 Nigeria Demographic and Health Survey (NDHS) which was implemented by the National Population Commission (NPC) with technical assistance provided by Inner City Fund (ICF) International through The Measure DHS Program [23]. Data collection was conducted from 14 August to 29 December 2018 with pre-test conducted from 30 April to 20 May 2018. The survey utilised a stratified two-stage sampling approach. In the first stage, 1389 enumeration areas (EAs) or clusters were selected as the primary sampling units. At the second stage, 40427 households were selected. Stratification was achieved by separating each administrative level 1 area (i.e., the 36 states and the Federal Capital Territory) into urban and rural areas, and samples were selected independently within each stratum. Detailed information on the description of the methods employed in the study is available elsewhere [23].

### Outcome variables

The primary outcome variables/indicators in this study are receipt of PENTA1 vaccine (n = 6059) and receipt of PENTA3 having received PENTA1 (PENTA3/1—the converse of dropout, n = 3937) among children aged 12–23 months, and receipt of MV (n = 11839) among children aged 12–35 months. For each of the three indicators, we assessed the binomial outcome: any evidence of vaccination versus no evidence of vaccination. For each of PENTA1 and MV, we additionally assessed the multinomial outcome: no evidence of vaccination, card invalid/history evidence of vaccination and card valid vaccination, to assess potential variation in associations that could be caused by misclassifying vaccination status when a verbal history of vaccination is accepted [24] (see Table A in S1 File). Valid and invalid vaccine doses were defined according to WHO guidance [25].

### Explanatory variables

**Demographic and Health Survey (DHS) covariates.**   The study considered covariates at the child-, household- and community-levels. The selection of these covariates was informed by literature on the predictors of vaccination coverage or of child health outcomes in general, expert opinion, and availability in the 2018 NDHS or other sources, as detailed in Table A in S1 File [6,11,17,21,26–35]. The study however excluded some pre-selected DHS covariates due to missingness or multicollinearity (see supplementary materials). These variables include preceding birth interval, antenatal and postnatal care, maternal receipt of tetanus toxoid vaccination (these variables had >15% missing cases for MV and were excluded from the analyses to make the results comparable across all the three indicators), maternal decision-making (whether mother decides health care, visits, and purchases)—and region of residence. Also, among similar covariates (e.g., mother's occupation and employment status), one was selected for inclusion in our model based on the literature and expert opinion.

**Geospatial covariates.**   The geospatial covariates retained in this study include travel time to health facility, enhanced vegetation Index (EVI) and livestock density index. Tertiles of the distribution of these covariate were used to allow similar number of observations for each tertile. We present further description about these covariates and their relevance in Fig A and Table B in S1 File [18–22,29,36,37]. Other geospatial covariates considered included distance to conflict locations, maximum number of conflicts, night light intensity, annual aridity index, maximum temperature, annual precipitation/rainfall, proximity to national borders, water, protected areas, and slope [18–20] were excluded from final models due to multicollinearity (as was EVI) or non-significance after adjusting for DHS variables.

## Data analysis

**Estimation of measures of association (fixed effects).** *Cross-tabulations and single-level logistic regression analysis.* We tabulated each outcome against each of the selected covariates separately to explore relationships and used Chi-squared tests to determine the significance of the associations. We then fitted frequentist single level simple logistic regression models to obtain the corresponding crude odds ratios (cORs) and associated 95% confidence intervals (CI). These results were later compared with results from the multiple multilevel analyses to determine changes in statistical significance and direction of effects.

*Multiple multilevel binomial regression analysis (any evidence of vaccination) and interaction effects.* We fitted Bayesian multilevel [38,39] binomial regression models to estimate adjusted odds ratios (aORs) and corresponding 95% credible intervals, accounting for the hierarchical structure of the data (child/household, community, and stratum levels) and, intrinsically, the survey design (clustering and stratification) through the last two hierarchies (see Fig B in S1 File). A detailed description of the model is included in the supplementary information (S1 File).

We investigated whether child/household covariate effects could be modified by the geospatial covariates by introducing interaction terms between both sets of covariates. To incorporate the interaction terms, we first fitted the main effects model using both DHS and geospatial covariates and then introduced the interactions between selected DHS and geospatial covariates sequentially, retaining only those that were significant in the final model.

*Multiple multilevel multinomial regression analysis (vaccination according to source of evidence).* The study also employed a Bayesian multinomial multivariable multilevel modelling approach to estimate the adjusted relative risk (aRR) and associated 95% credible intervals for covariates significantly associated with PENTA1 and MV using the multinomial outcomes defined previously. No interaction terms were considered for the multinomial analyses due to model complexities and non-convergence challenges.

## Measures of variation (random effects analysis)

We computed summary measures [30,40] of the amount of residual variation attributable to the hierarchies in the binomial models. These included the variance partitioning coefficient (VPC) which measures residual variation between clusters/communities in different strata; the median odds ratio (MOR) which quantifies residual community level variation in the likelihood of vaccination on the odds ratio scale, and the percent change in variance (PCV) which measures change in residual variation due to the inclusion of covariates in the models [41–45].

Also, although prediction was not the main goal, we evaluated the discriminatory or predictive power of the fitted models using the area under the receiver operating characteristic (AUROC) curve (see supplementary materials for details).

All analyses were implemented in Stata version 16 [46], MLwiN version 3.05 [47], and the R programming language version 4.0.3 [48]. Additionally, we used the runmlwin [49] program to run the MLwiN multilevel modelling software from within Stata. We utilized MCMC algorithms with a burn-in length of 1000, a monitoring chain length of 60000, and thinning of 20. Convergence of the MCMC chains was assessed via visual inspection of the trace and autocorrelation plots of the parameters.

## Ethical approval and consent to participate

Ethical approval was obtained from the Nigeria National Health Research Ethics Committee and the ICF Institutional Review Board for the main NDHS [23], and from the Ethics and Research Governance, University of Southampton, United Kingdom. Written informed

consent was obtained from all study respondents. However, the data were analysed anonymously in the present study. All methods were performed in accordance with the relevant guidelines and regulations.

## Results

### Outcome indicators of vaccination coverage in Nigeria

Among 6059 children aged 12–23 months, 3937 (65%) had any evidence of receiving PENTA1 and 3041 PENTA3 vaccination hence PENTA3/1 was 77%. Among the 2039 children with documented evidence of PENTA1 vaccination at or after 6 weeks of age, 1769(86.7%) had documented receipt of PENTA3 vaccine, while PENTA3/1 was lower (67.8%) among the children with only a verbal history of vaccination. Among 11839 children aged 12–35 months, 2111 (18%) had documented receipt of a valid dose of MV (at or after age 9 months); 4522 (38%) had either invalid documented doses or a verbal history of vaccination, and 5206 (44%) had no evidence of MCV receipt (see Tables C, F, H, J, L in S1 File).

### Cross-tabulation results

As expected, the receipt of PENTA1, PENTA3/1 and MV was higher among children with a health card/document (or a home-based record—HBR) than those without. Multiple other individual and family factors related to socio-economic and demographic status, access to communications technology, use of other services and length of stay in household were significantly associated with at least one of the outcome indicators using Chi-squared tests. Of geospatial/community variables, rural/urban residence, livestock density index, travel time to nearest health facility, and vegetation index were associated with all three indicators (see Tables C, F and H in S1 File).

### Multiple multilevel binomial analyses results

Figs 1–3 show the adjusted odds ratios and their corresponding 95% credible intervals for receipt of PENTA1, PENTA3/1 and MV vaccination, based on any evidence of vaccination—definitions and reference categories are described in Table A in S1 File and detailed results shown in Tables D, E, G and I in S1 File. Note that Fig 1 shows the results for both the main effects and the interaction terms for PENTA1.

Factors associated with vaccination were broadly similar for documented versus recall evidence of vaccination and included individual and community (geospatial) attributes. Although coverage of each vaccine-dose was higher in urban than rural areas, urban status was not significant in multivariable analyses. Based on any evidence of vaccination, we found that HBR, receipt of vitamin A and higher maternal educational level were significantly positively associated with each outcome. Indicators relating to socio-economic status, as well as ethnic group, skilled birth attendance, lower travel time to a clinic and reported problems seeking health care were significantly associated with both PENTA1 and MV. Maternal religion was related to PENTA1, and PENTA1/3 and maternal age related to MV and PENTA3/1; other variables were significantly associated with one outcome each.

There were few differences in determinants of receipt of PENTA1 compared to MV (Fig 1 and Figs 3 and 4).

Being Christian (aOR = 1.53, 95% Cr.I: 1.08, 2.10, compared with families practising Islam), having a length of stay of < 1 year (aOR = 2.32, 95% Cr.I: 1.10, 4.29) or > 5 years (aOR = 1.53, 95% Cr.I: 1.05, 2.17), and residence in communities with lower livestock density index (aOR = 1.66, 95% Cr.I: 1.13, 2.37, reference higher livestock index) or those with medium vegetation index (aOR = 1.45, 95% Cr.I: 1.07, 1.93, reference lower vegetation index)

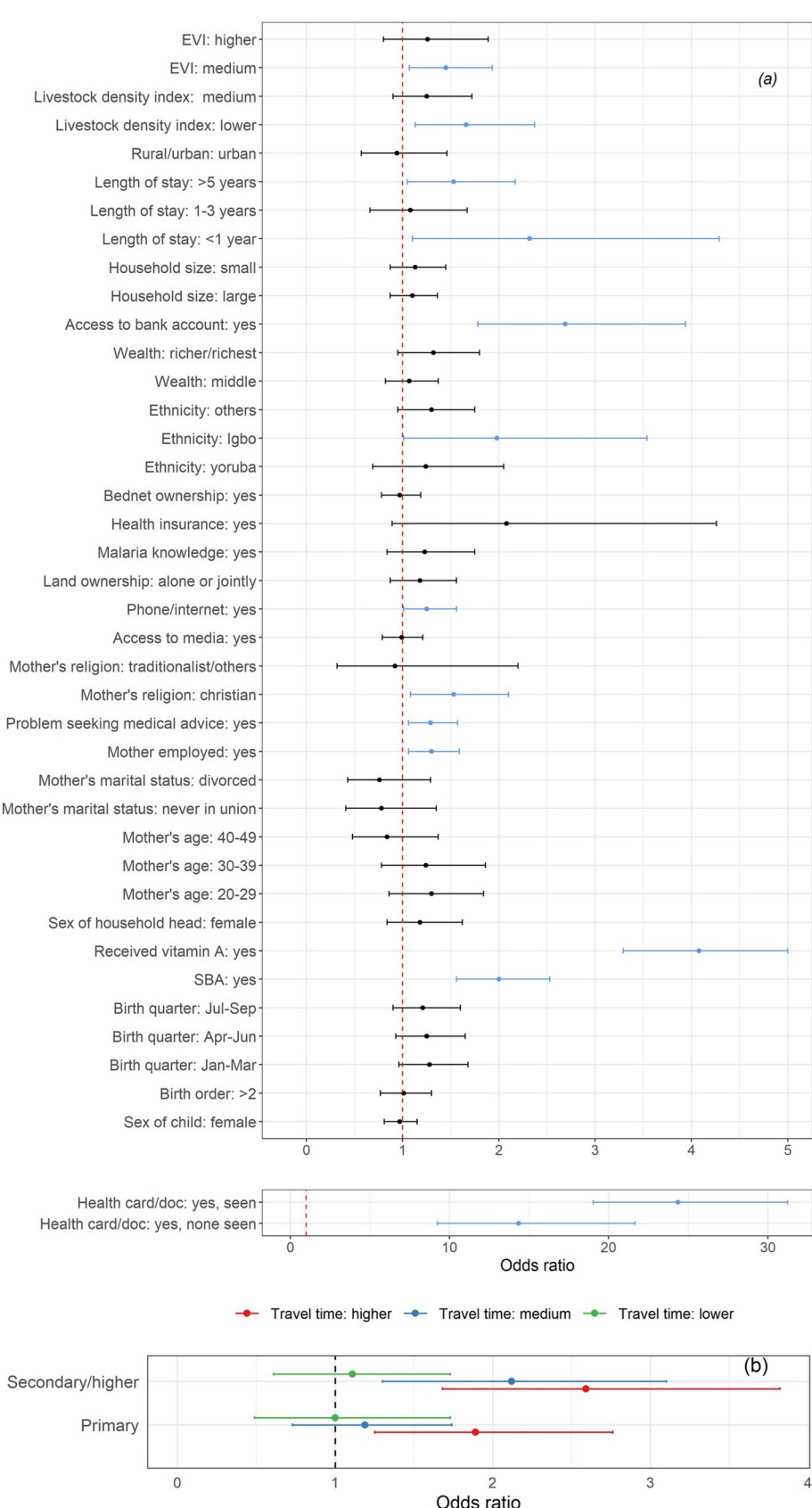

**Fig 1.** (a) Plots of adjusted odds ratios and corresponding 95% credible intervals based on the Bayesian multilevel binomial analysis for PENTA1. The vertical dotted lines mark the odds ratio of 1 and the blue lines mark significant covariates. (b) Interaction effects between maternal education and travel time to the nearest health facility for PENTA1. The vertical dotted line marks the odds ratio of 1.

were associated with higher odds of receipt of PENTA1 but not MV. Significant interaction between maternal education and travel time to the nearest health facility was found in the model for PENTA1 vaccination (see Fig 1(B), and Tables D and E in S1 File). The effect of education was greatest among those with higher travel times and was not significant among the lowest travel times, while the effect of longer travel times was only significant among those whose mothers had no education (see Table E in S1 File).

Birth order, wealth, health insurance, bednet ownership and access to traditional media were significantly associated with MV receipt but not the other outcomes. Interestingly, reporting a problem seeking health care was positively associated with receipt of both PENTA1 and MV. Among those who received PENTA1, receipt of PENTA3 was associated with mother's age, education, and knowledge of malaria, HBR possession and receipt of vitamin A (see Fig 2, and Table G in S1 File).

Finally, we summarized significant covariates in Fig 4 for easy reference.

Substantial residual community level variances in different strata were observed in the fitted model for each outcome (Tables D, G and I in S1 File). Specifically, 26%, 15%, and 19% of variation in PENTA1, PENTA3/1 and MV, respectively, could be attributable to communities in different strata. The median odds ratios of 2.7, 2.1, and 2.3 respectively for PENTA1, PENTA3/1 and MV, which are at least two times higher than the reference value (MOR = 1), are an indication of substantial community level variation in the likelihood of receiving PENTA1, PENTA3/1 and MV vaccinations. Also, the estimated PCV values demonstrate that the covariates included in our models led to 67%, 61%, and 57% reduction in residual variation at both the community and stratum levels in the fitted models for PENTA1, PENTA3/1 and MV respectively. Lastly, the discriminatory or predictive power of the fitted models for correctly predicting the likelihood of PENTA1, PENTA3/1, and MV vaccinations based on the AUROC curve were 91.3%, 77.1% and 80.2% respectively as displayed in Fig C in S1 File.

## Multiple multilevel multinomial analyses results

Overall, there were few differences in the direction and magnitude of associations between the independent variables and the outcomes classified according to source of evidence of vaccination ("card valid" or "card invalid/history")–Fig 5, and Fig D and Tables J-M in S1 File. A few differences were found, however. For example, for PENTA1, lower travel time significantly increased the odds of card valid vaccination but not card invalid/history vaccination and the effect of receipt of vitamin A was highest for card valid vaccination. For MV, areas with lower travel time (compared to higher travel time) and middle wealth status (compared to poorer/ poorest) had significantly higher likelihood of card valid vaccination (relative to no evidence of vaccination) but not card invalid/history vaccination. Receipt of vitamin A was significantly associated with MV vaccination irrespective of source of evidence, but the effect was greater for card valid MV vaccination. Length of stay less than 1 year was associated with card invalid/ history of vaccination but not card valid vaccination.

## Discussion

Our analyses of correlates of vaccination include several innovations. First, we examined separately factors associated with beginning (PENTA1) and completing (PENTA3/1 and MV) the

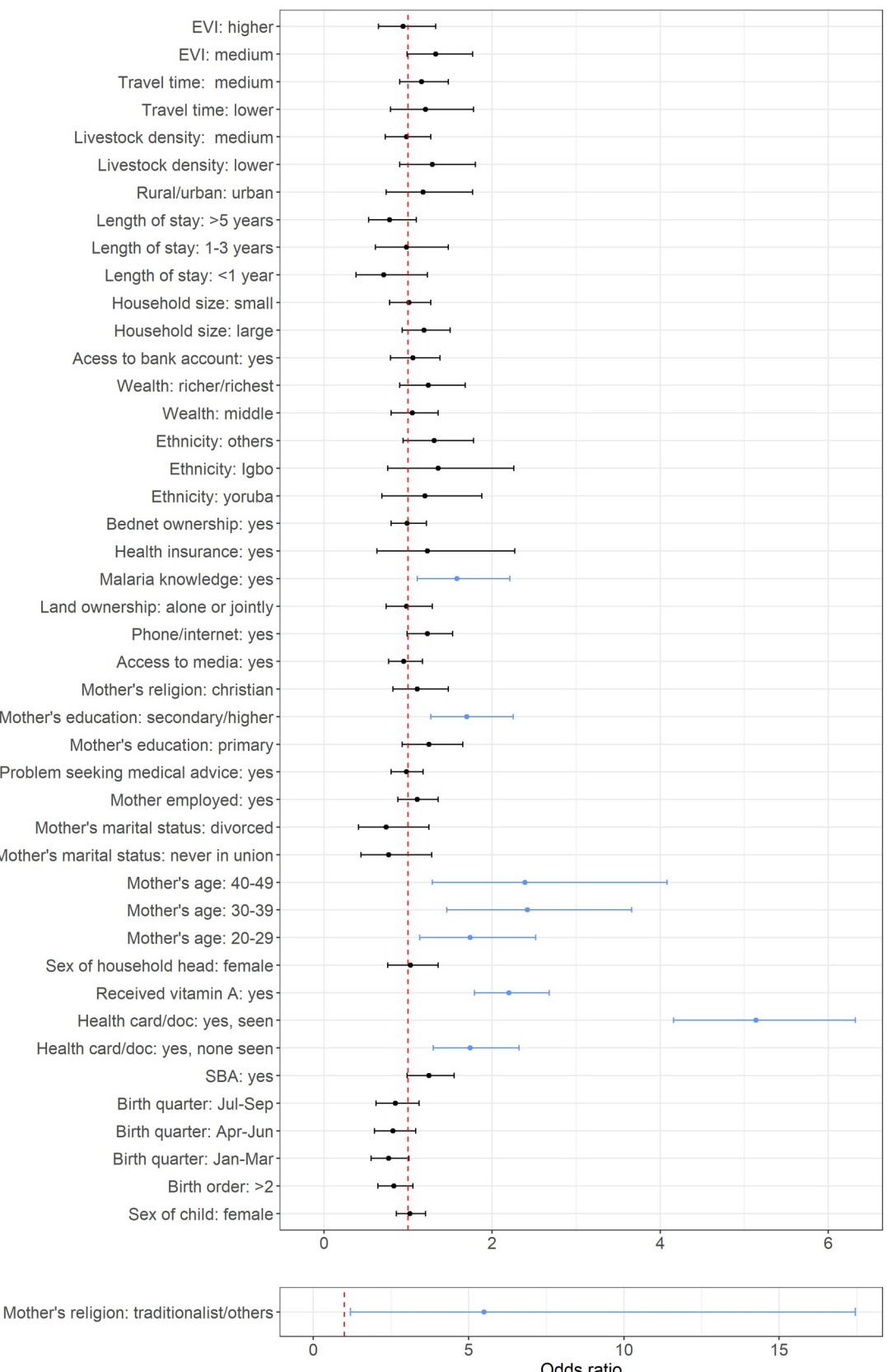

**Fig 2. Plots of adjusted odds ratios and corresponding 95% credible interval based on Bayesian multilevel binomial analysis for PENTA3/1 (completion of the PENTA series among those who received PENTA1).** The vertical dotted line marks the odds ratio of 1 and the blue lines mark significant covariates.

vaccination series. Second, we included community-level geospatial factors as well as individual attributes. Third, we used multinomial analyses to examine potential differences in findings when analyses were stratified by the strength of evidence of vaccination. Differences between associations of receipt of PENTA1 according to source of evidence may relate to misclassification of the outcome when a verbal history is accepted. The fact that we found few differences suggests that the mother's recall was reasonably accurate, as reported elsewhere [24,50]. For MV, differences according to source of evidence may also relate to the vaccination strategy used. Children aged 12–35 months in Aug-Dec 2018 (when the DHS fieldwork was done) were mostly eligible for the November 2017 measles campaign in Nigeria. Doses recorded on the HBR represent only those received via routine immunization while a verbal history may include doses received during campaigns although we have previously found evidence that these are under-reported in Nigeria [20].

Overall, our analyses showed that for both PENTA1 and MV, factors that were predictive of card invalid/history vaccination were broadly similar to those that were predictive of card valid vaccination (i.e., the multinomial analysis), and to those found when considering any evidence of vaccination (i.e., the binomial analysis). There was a difference for travel time, where for both PENTA1 and MV, lower travel time to a health facility significantly increased the odds of card valid vaccination but not card invalid/history evidence of vaccination, indicating that proximity to a health facility has a marked influence on the timeliness and validity of vaccinations [4,21,51]. In what follows, we focus on the factors identified in the binomial analyses.

Ownership of a health card/document, receipt of Vitamin A—both of which are indicators of access to health services, and maternal education were positively associated with all three coverage indicators, the effect being modified by travel time for PENTA1, with mothers lacking education in remote areas being least likely to attend for vaccination. Skilled attendance at birth had significant positive associations with both PENTA1 and MV, further highlighting the importance of access to health services in improving vaccination coverage and corroborating findings in previous studies [5,10,11,15,16,28,34,35,52–55]. Wealth-related indicators were associated with both PENTA1 and MV although some specific indicators differed, for example for PENTA1, maternal employment and having a bank account were important while for MV, wealth, health insurance, and bednet ownership were important. As is commonly found, economically empowered mothers make better health choices for their children [56,57]. The association of PENTA1 with access to a mobile phone/internet may reflect both wealth and access to health information. Children born to Igbo mothers compared to Hausa/Fulani and those born to Christian mothers compared to Muslim mothers were more likely to receive PENTA1, which contributes to the geographical disparities in routine immunization (RI) coverage in the country [58,59]. Associations of PENTA1 with livestock density and vegetation index may also reflect geographical disparities. Interestingly, the likelihood of PENTA1 receipt was higher among children born to mothers who reported a problem seeking medical advice/treatment, which we speculate to be a consequence of higher motivation and better health-seeking behaviour among these women. Thus, interventions seeking to improve vaccination coverage in remote areas should be designed especially with mother's educational level in mind [10,11,16].

While socio-economic factors helped predict PENTA1 receipt, among those who started the PENTA series, mother's education, knowledge of malaria, being at least ≥ 20 years old, possibly an indication of better knowledge of vaccination [16,17,28,34,52,60,61] and practising

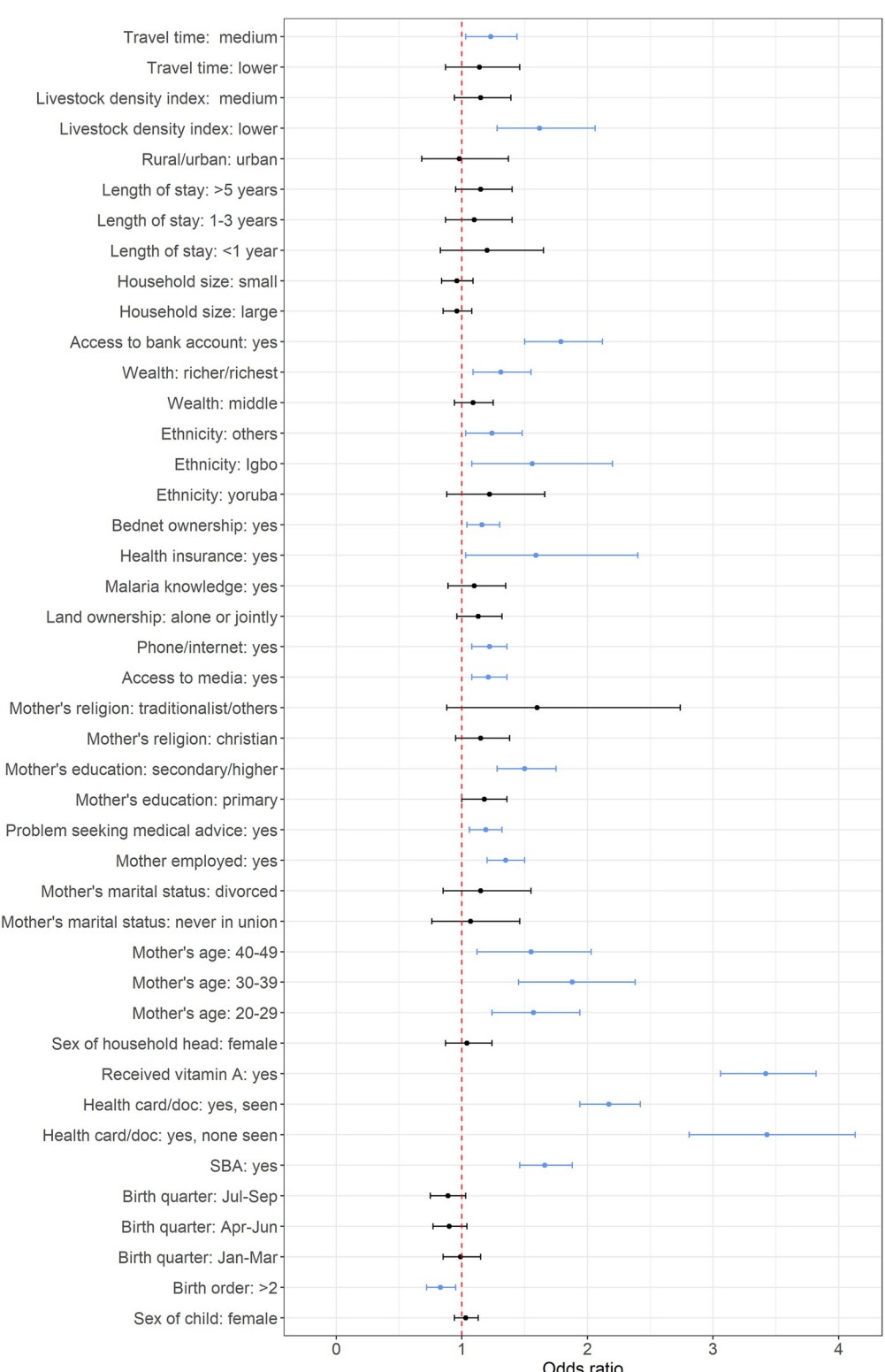

**Fig 3. Plots of adjusted odds ratios and corresponding 95% credible interval based on Bayesian multilevel binomial analysis for MV vaccination coverage.** The vertical dotted line marks the odds ratio of 1 and the blue lines mark significant covariates.

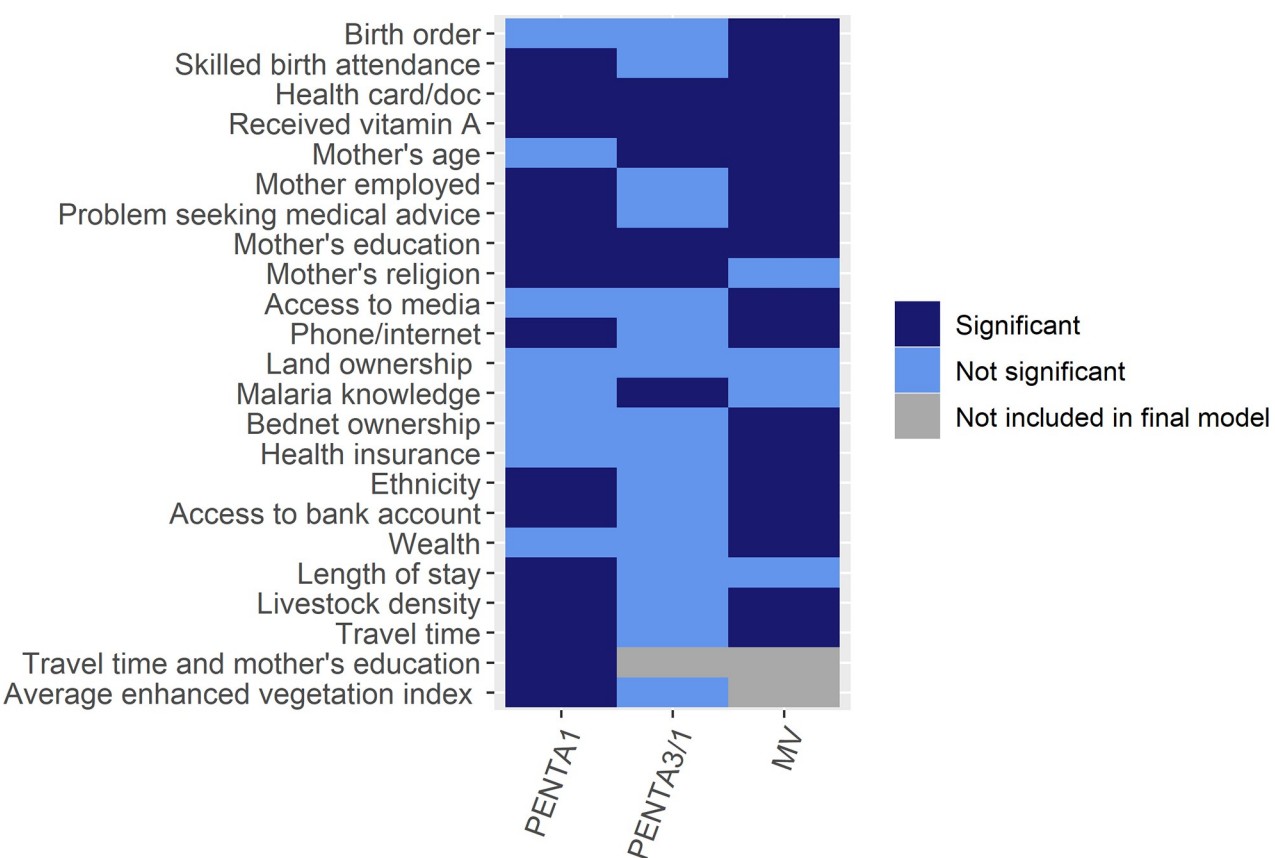

**Fig 4. Summary of factors predictive of PENTA1, PENTA3/1 and MV vaccinations including the interaction term for PENTA1.**

traditional/other religion (compared to Islam) were associated with increased likelihood of completion of the three-dose PENTA series (i.e. PENTA3/1), further highlighting the importance of maternal education and knowledge.

MV receipt was related both to socio-economic variables and access to media, birth order, health insurance, and bednet ownership. Children who had a birth order above two were less likely to receive MV, signalling that even though older mothers were more likely to have their children vaccinated, this propensity could change with subsequent births [13,14,16,58,62,63]. Health insurance and bednet ownership could reflect both wealth and a positive attitude to health care.

Finally, our analyses revealed substantial variation in the likelihood of vaccination at the community level, demonstrating the need for estimation of coverage and targeting of interventions at granular spatial scales [18,19].

## Study limitations

First, the study could not establish causal relationships due to the cross-sectional sampling design. Secondly, our study did not comprehensively assess all the factors that could affect vaccination coverage, particularly attitudes towards vaccination and supply-side factors such as vaccine and health worker availability and missed opportunities for vaccination [64] due to data limitations. Some important covariates such as antenatal care, postnatal care, and mother's receipt of tetanus toxoid injections before birth were excluded from the analysis because they had > 15% missing data for MV, but these would likely correlate with skilled birth

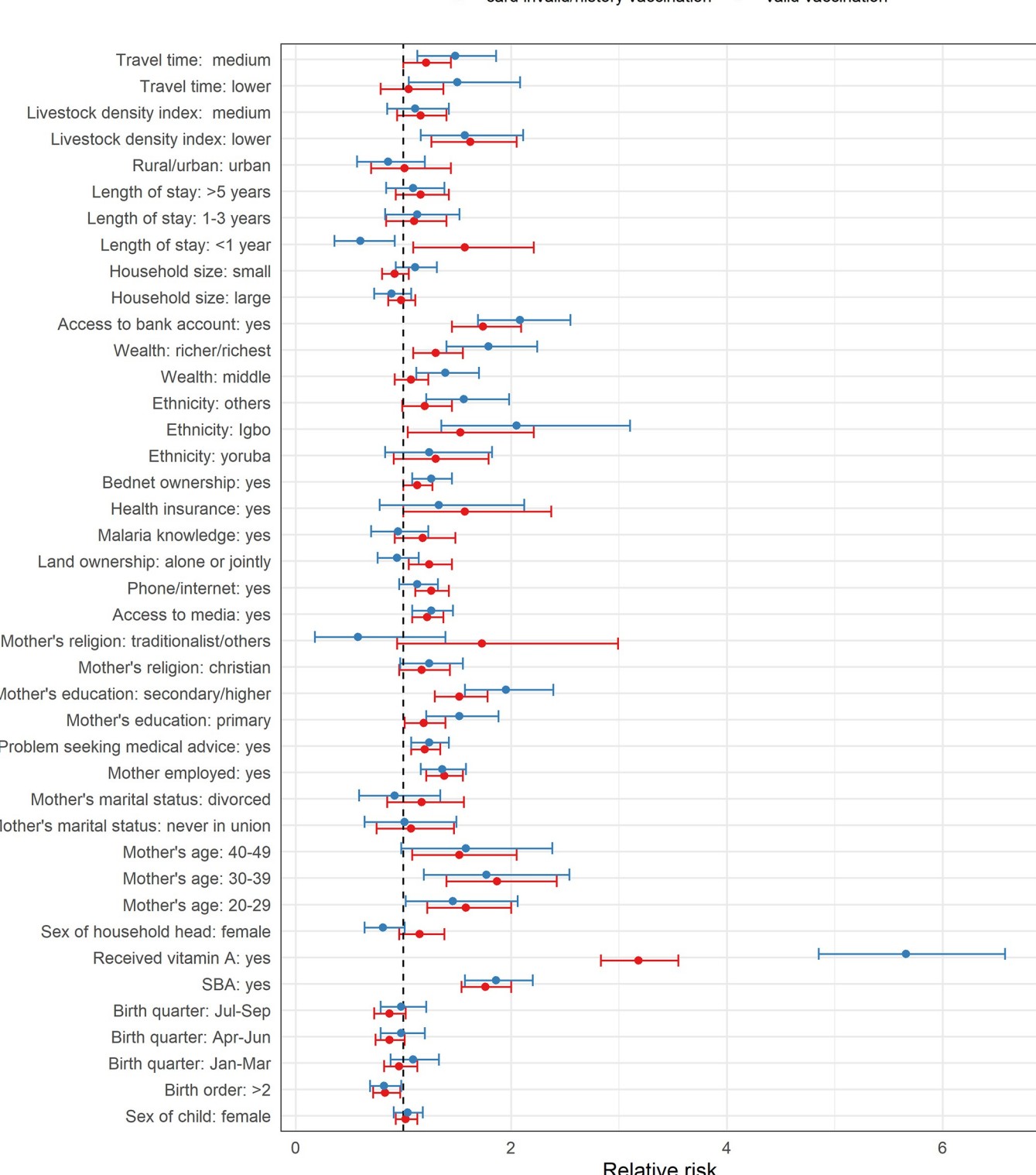

**Fig 5. Plots of relative risks and corresponding 95% credible interval based on Bayesian multilevel multinomial analysis for MV vaccination coverage.** The vertical dotted line marks the relative risk of 1.

attendance. Lastly, our analysis may have excluded important at-risk populations such as those living in conflict-affected areas and urban slums if the sampling frame used in the DHS did not fully capture these populations.

## Conclusion

The study has identified several factors to be predictive of indicators of childhood vaccination coverage pointing to the need for an integrated approach to addressing inequities in vaccination coverage in Nigeria. This should include improvements in access to health facilities and services (e.g., skilled birth attendance), socioeconomic conditions of households, and improvements in maternal education through targeting uneducated and teenage mothers with health literacy programmes including familiarization with the vaccination schedule and the importance of retention of home-based records. Also, better utilization of means of communication such as the traditional media and mobile phones/internet for disseminating vital health information is likely to yield improvements in coverage. Furthermore, community-focused interventions, and further research will be required to identify other supply- and demand-side factors as part of an overall strategy to improve childhood vaccination coverage in Nigeria. Also, the effects of the geospatial covariates are estimated for the entire country in our models. It will make sense to display maps of these if they were estimated for each region, for example. We will explore this detailed analysis in our future work.

## Supporting information

**S1 File.** Supplementary file containing Tables A–M, Figs A-D and additional texts referenced in the manuscript.
(DOCX)

## Author Contributions

**Conceptualization:** Andrew J. Tatem, Felicity T. Cutts, C. Edson Utazi.

**Data curation:** Justice Moses K. Aheto, Oliver Pannell, Winfred Dotse-Gborgbortsi, Mary K. Trimner, Dale A. Rhoda, C. Edson Utazi.

**Formal analysis:** Justice Moses K. Aheto, Winfred Dotse-Gborgbortsi, Dale A. Rhoda, Felicity T. Cutts, C. Edson Utazi.

**Funding acquisition:** C. Edson Utazi.

**Investigation:** Justice Moses K. Aheto, C. Edson Utazi.

**Methodology:** Justice Moses K. Aheto, C. Edson Utazi.

**Project administration:** Andrew J. Tatem, C. Edson Utazi.

**Resources:** Andrew J. Tatem.

**Software:** Justice Moses K. Aheto.

**Supervision:** Andrew J. Tatem.

**Validation:** Justice Moses K. Aheto, C. Edson Utazi.

**Visualization:** Justice Moses K. Aheto.

**Writing – original draft:** Justice Moses K. Aheto, Felicity T. Cutts, C. Edson Utazi.

**Writing – review & editing:** Justice Moses K. Aheto, Oliver Pannell, Winfred Dotse-Gborg-bortsi, Mary K. Trimner, Andrew J. Tatem, Dale A. Rhoda, Felicity T. Cutts, C. Edson Utazi.

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
