## [Decision Letter · Decision Letter 0]

22 Nov 2021

PONE-D-21-28231Multilevel analysis of predictors of multiple indicators of childhood vaccination in NigeriaPLOS ONE

Dear Dr. Aheto,

Thank you for submitting your manuscript to PLOS ONE. After careful consideration, we feel that it has merit but does not fully meet PLOS ONE’s publication criteria as it currently stands. Therefore, we invite you to submit a revised version of the manuscript that addresses the points raised during the review process.

We look forward to receiving your revised manuscript.

Kind regards,

Tzai-Hung Wen, Ph.D.

Academic Editor

PLOS ONE

Journal Requirements:

2. Thank you for stating the following in the Acknowledgments/ Funding Section of your manuscript: 

This work was supported by funding from the Bill and Melinda Gates Foundation (Investment ID INV-003287). CEU received the grant. The funder did not play any role in the study design, data collection, analysis and interpretation of data, the report writing, and the decision to submit the manuscript for publication. 

This work was supported by funding from the Bill and Melinda Gates Foundation (Investment ID INV-003287). CEU received the grant. The funder did not play any role in the study design, data collection, analysis and interpretation of data, the report writing, and the decision to submit the manuscript for publication. 

5. We note that Figure S1 in your submission contain [map/satellite] images which may be copyrighted. All PLOS content is published under the Creative Commons Attribution License (CC BY 4.0), which means that the manuscript, images, and Supporting Information files will be freely available online, and any third party is permitted to access, download, copy, distribute, and use these materials in any way, even commercially, with proper attribution. For these reasons, we cannot publish previously copyrighted maps or satellite images created using proprietary data, such as Google software (Google Maps, Street View, and Earth). For more information, see our copyright guidelines: http://journals.plos.org/plosone/s/licenses-and-copyright.

a. You may seek permission from the original copyright holder of Figure S1 to publish the content specifically under the CC BY 4.0 license.  

Reviewers' comments:

Reviewer's Responses to Questions

**Comments to the Author**

1. Is the manuscript technically sound, and do the data support the conclusions?

Reviewer #1: Yes

Reviewer #2: Yes

2. Has the statistical analysis been performed appropriately and rigorously? 

Reviewer #1: Yes

Reviewer #2: Yes

3. Have the authors made all data underlying the findings in their manuscript fully available?

Reviewer #1: Yes

Reviewer #2: Yes

4. Is the manuscript presented in an intelligible fashion and written in standard English?

Reviewer #1: Yes

Reviewer #2: Yes

5. Review Comments to the Author

Reviewer #1: Overall comment:

This study investigates multiple indicators of childhood vaccination in Nigeria. Using data from the 2018 Nigeria Demographic and Health Survey and geospatial data sets, authors built Bayesian multilevel binomial and multinomial logistic regression models to analyze independent predictors of three vaccination outcomes. Although authors claimed several innovations, including they examined separately factors associated with beginning (PENTA1) and completing (PENTA3/1 and MV) the vaccination series; included community-level geospatial factors as well as individual attributes, and used multinomial analyses to examine potential differences in findings when analysis, I think this study has normal quality. A major revision can be reached. The model design and the description of results are good, but there are still some issues that should be improved before publish.

1. Line 66-68. The major finding is not so attractive for me. Because it is basically a piece of general knowledge, we can easily to imagine the result.

2. Line 92-114: Although the motivation and objective are clear, the quality of the introduction is improvable. There must be several relative studies that use a similar approach to investigate the problem, but the authors didn’t mention about it.

3. Line 104-106: More than ten cited references are included in only one sentence. Authors are suggested to address more detail about them.

4. Line 155: I think authors include the geospatial covariates in models is a very good approach. But I suggest authors can present the geospatial covariates by maps.

5. Line 175-211: The model design is praiseworthy.

6. Line 308-313: Similarly, I suggest authors can display the results of the effect of geospatial factors by maps.

7. Figures: The quality and resolution of figures are very bad. It’s hard to know the results.

8. Tables: I think some numerical statistical findings can be presented as tables.

Reviewer #2: Review of PLUS ONE

PONE-D-21-28231: " Multilevel analysis of predictors of multiple indicators of childhood vaccination in Nigeria "

The manuscript presents a reasonably well-articulated discussion for the developed framework to address childhood vaccination inequalities in Nigeria. Based on my reading, the applications of multilevel logistic models to produce new insights to data analysis makes the current manuscript contributing to the literature of public health. The work will be of also interest to a wide readership in the journal. Although the manuscript is worth published, its present form needs some revisions and I have the following suggestions.

1. In the Introduction, the authors omitted the brief description about multilevel models. Some details can be found at supplementary file but I think the associated discussions could be summarized into the Introduction. In so doing, the article would be more complete and clear.

2. In the Data section, the authors may provide a figure for the hieratical structure of the data, which can furnish the rationale why to use multilevel models.

On the other hand, for readability and the ease of understanding, the authors should at least include some model equations for the multilevel models in the section of Data analysis, rather than just assemble all the materials in the supplementary.

3. Is there any specific reason to use Bayesian multilevel models? As the ‘frequentist’ multilevel logistic regression models exist as alternative ways to analyze the data, I suggest the authors explain a bit what to motivate the Bayesian multilevel analyses and why to use them both in the Introduction and Data analysis sections.

4. Regarding the model specifications, do the multilevel models only include random intercepts? Is there any random coefficients for the predictors? From the model equation 2 and equation 3 in the supplementary file, it is not clear to me how the multilevel representation is given.

5. The authors indicated in the supplementary that non-informative priors can be alternative choice of priors for the model parameters. I was actually thinking that would it be possible to generate better analysis results as there is lack of information on reliable priors. Although I don‘t demand the authors to do so, the discussion needs to be added.

6. Lines 168-174: It is not clear whether the single-level logistic regression analysis was taken with Bayesian approach or not. This needs to be clarified.

7. Lines: 190-194: The authors did not consider interaction terms in the multinomial multilevel regression analysis. I wonder if this is due to the computational difficulty or convergence problems for the considered models? Also, does it need proportional odds assumption here? The authors may discuss these in the manuscript.

6. PLOS authors have the option to publish the peer review history of their article (what does this mean?). If published, this will include your full peer review and any attached files.

Reviewer #1: No

Reviewer #2: No

---

## [Author Response · Author response to Decision Letter 0]

22 Dec 2021

Dear Editor,

Thank you for the opportunity to once again revise our manuscript titled “Multilevel analysis of predictors of multiple indicators of childhood vaccination in Nigeria”. We are extremely grateful to the Editors and the Reviewers for the constructive criticism of the manuscript and the positive feedback which certainly helped to improve the message and the quality of our manuscript. We are indeed grateful.

We have duly addressed all the concerns raised by the Editors and the Reviewers to the best of our ability and wish to re-submit the revised version for your consideration and subsequent publication in your cherished journal, and together we can all help in addressing this serious public health challenge facing millions of children globally, especially in developing countries like Nigeria. 

Response to Editorial comments

Response: We have revised the manuscript to address the issues about the style requirements of the journal throughout the manuscript text. 

2. Thank you for stating the following in the Acknowledgments/ Funding Section of your manuscript: 

This work was supported by funding from the Bill and Melinda Gates Foundation (Investment ID INV-003287). CEU received the grant. The funder did not play any role in the study design, data collection, analysis and interpretation of data, the report writing, and the decision to submit the manuscript for publication. 

This work was supported by funding from the Bill and Melinda Gates Foundation (Investment ID INV-003287). CEU received the grant. The funder did not play any role in the study design, data collection, analysis and interpretation of data, the report writing, and the decision to submit the manuscript for publication. 

Response: We removed the funding information under the Funding section of the manuscript and stated this as “This work was supported by funding from the Bill and Melinda Gates Foundation (Investment ID INV-003287). CEU and AJT received the grant. The funder did not play any role in the study design, data collection, analysis and interpretation of data, the report writing, and the decision to submit the manuscript for publication” in the cover letter as directed. The Editors should use this for the publication. 

Response: Thank you for this information. We revised the corresponding author’s affiliation to reflect this. The corresponding author’s address and affiliations are as follows in order as they appear:

1) WorldPop, School of Geography and Environmental Science, University of Southampton, Southampton, SO17 1BJ, UK.

2) Department of Biostatistics, School of Public Health, College of Health Sciences, University of Ghana, Accra - Ghana”.

We have revised the affiliations of the other authors accordingly to reflect the changes made. (see the title page). 

Response: All the data sets supporting the analyses in this manuscript are publicly available and can be obtained by sending data request to the owners or by accessing them from the owner’s website. We provided full details on the data sources, and references to enable readers to access these data. We are unable to directly share this data with third parties because we did not have permission from the data owners to do so. We have provided additional text as “The authors did not have any special access privileges that others would not have” to the data availability section. 

Response: All the data (both DHS and Geospatial variables) supporting the analyses presented in this manuscript are publicly available and sources provided in the methods section. In addition, we presented the description, the sources, and the references to the geospatial covariates used in our work in Table S2, and also under “Geospatial covariate description and processing steps” in the Supplementary Information. 

5. We note that Figure S1 in your submission contain [map/satellite] images which may be copyrighted. All PLOS content is published under the Creative Commons Attribution License (CC BY 4.0), which means that the manuscript, images, and Supporting Information files will be freely available online, and any third party is permitted to access, download, copy, distribute, and use these materials in any way, even commercially, with proper attribution. For these reasons, we cannot publish previously copyrighted maps or satellite images created using proprietary data, such as Google software (Google Maps, Street View, and Earth). For more information, see our copyright guidelines: http://journals.plos.org/plosone/s/licenses-and-copyright.

a. You may seek permission from the original copyright holder of Figure S1 to publish the content specifically under the CC BY 4.0 license. 

Response: The Figure S1 was produced by the authors for this manuscript. Thus, we did not need any permission to use it, and so did not make any changes to the manuscript.

Response: The maps and its contents presented in Figure S1 in the Supplementary Information was produced by the authors using publicly available data in ArcGIS software for this paper, and the data source were cited in Table S2 in the Supporting Information. We did not need any permission to use this figure as it was produced by us. We have published similar maps in previous studies[1-3]. Hence, we did not make any changes to the manuscript. 

Reference

1. Utazi CE, Thorley J, Alegana VA, Ferrari MJ, Takahashi S, Metcalf CJE, et al. Mapping vaccination coverage to explore the effects of delivery mechanisms and inform vaccination strategies. Nature Communications. 2019;10(1):1633. doi: 10.1038/s41467-019-09611-1.

2. Utazi CE, Wagai J, Pannell O, Cutts FT, Rhoda DA, Ferrari MJ, et al. Geospatial variation in measles vaccine coverage through routine and campaign strategies in Nigeria: Analysis of recent household surveys. Vaccine. 2020;38(14):3062-71. doi: https://doi.org/10.1016/j.vaccine.2020.02.070.

3. Utazi CE, Thorley J, Alegana VA, Ferrari MJ, Takahashi S, Metcalf CJE, et al. High resolution age-structured mapping of childhood vaccination coverage in low and middle income countries. Vaccine. 2018;36(12):1583-91. Epub 2018/02/14. doi: 10.1016/j.vaccine.2018.02.020. PubMed PMID: 29454519; PubMed Central PMCID: PMCPMC6344781.

Response: We revised the manuscript to reflect these changes. The marked changes can be found after the reference list under the heading Supporting Information captions. 

Reviewers' comments:

Reviewer's Responses to Questions

Comments to the Author

1. Is the manuscript technically sound, and do the data support the conclusions?

Reviewer #1: Yes

Reviewer #2: Yes

2. Has the statistical analysis been performed appropriately and rigorously?

Reviewer #1: Yes

Reviewer #2: Yes

3. Have the authors made all data underlying the findings in their manuscript fully available?

Reviewer #1: Yes

Reviewer #2: Yes

4. Is the manuscript presented in an intelligible fashion and written in standard English?

Reviewer #1: Yes

Reviewer #2: Yes

5. Review Comments to the Author

Reviewer #1: Overall comment:

This study investigates multiple indicators of childhood vaccination in Nigeria. Using data from the 2018 Nigeria Demographic and Health Survey and geospatial data sets, authors built Bayesian multilevel binomial and multinomial logistic regression models to analyze independent predictors of three vaccination outcomes. Although authors claimed several innovations, including they examined separately factors associated with beginning (PENTA1) and completing (PENTA3/1 and MV) the vaccination series; included community-level geospatial factors as well as individual attributes, and used multinomial analyses to examine potential differences in findings when analysis, I think this study has normal quality. A major revision can be reached. The model design and the description of results are good, but there are still some issues that should be improved before publish.

Response: Thank you for the positive and helpful comments.

1. Line 66-68. The major finding is not so attractive for me. Because it is basically a piece of general knowledge, we can easily to imagine the result.

Response: Thank you. To our knowledge, our study is the first of its kind to examine the relationships between vaccination and a range of predictor variables based on strength of evidence of vaccination, i.e., documented versus recall evidence of vaccination. We consider our finding that “factors associated with vaccination were broadly similar for documented versus recall evidence of vaccination” to be highly significant as this will help guide policymakers when making decisions using different evidence of vaccination available to them. Moreover, our findings are important because we were also interested in establishing which predictor might be associated with all the three outcomes (Penta1, Penta3/1, and MV) under investigation. Hence, policies addressing the common predictors of all three outcomes is likely to generate greater improvements in coverage levels. 

2. Line 92-114: Although the motivation and objective are clear, the quality of the introduction is improvable. There must be several relative studies that use a similar approach to investigate the problem, but the authors didn’t mention about it.

Response: Thank you. We revised the manuscript to reflect this. See line 122, and lines 125-138 at page 4. 

3. Line 104-106: More than ten cited references are included in only one sentence. Authors are suggested to address more detail about them.

Response: Thank you. We revised the manuscript to expatiate on this. See lines 115-121 at pages 3 and 4. 

4. Line 155: I think authors include the geospatial covariates in models is a very good approach. But I suggest authors can present the geospatial covariates by maps.

Response: Thank you for the positive feedback. We have already provided the maps for the geospatial covariates used in our models in Figure S1 in the Supplementary Information at page 6. 

5. Line 175-211: The model design is praiseworthy.

Response: Thank you. We appreciate this feedback. 

6. Line 308-313: Similarly, I suggest authors can display the results of the effect of geospatial factors by maps.

Response: Thank you. The effects of the geospatial covariates are estimated for the entire country in our models. It will make sense to display maps of these if they were estimated for each region, for example. We will explore this detailed analysis in future work as we have now highlighted in the discussion (see lines 440-443 at page 17). However, we mapped the values of all the geospatial covariates retained in our models across the entire Nigeria to help the readers appreciate and understand the spatial variation in these covariates. See Figure S1 in the Supplementary Information at page 6. 

7. Figures: The quality and resolution of figures are very bad. It’s hard to know the results.

Response: Thank you. The figures as appeared on our system as .tiff file are all clear when you open them. Perhaps, what you observed might be due to the online system combining all the files as PDF. Note that we used the ‘Preflight Analysis and Conversion Engine (PACE) digital diagnostic tool recommended by PLOS ONE and available at https://pacev2.apexcovantage.com/ to prepare our figures to the acceptable format and the standard required by PLOS ONE. Also, we saved these figures in R software directly as .tiff files and then used the PACE tool to prepare them as required by the journal, and all the figures were approved as meeting the PLOS ONE standard by the PACE system. 

8. Tables: I think some numerical statistical findings can be presented as tables.

Response: Thank you. Given the numerous numbers of parameters involved in our models, it is preferable to summarise the results in figures instead of tables for the main manuscript text to sustain readers interests while the tables for the same results are sent to the Supplementary Information. It is also easier to understand the results as presented in the figures than in tables while maintaining the scientific meaning of these results. However, note that we have already provided same results in the supplementary tables in the Supplementary Information for those who might also be interested in reading these results in tables (see Tables S3-S13). Thus, we did not make any changes to the manuscript. 

Reviewer #2: Review of PLUS ONE

PONE-D-21-28231: " Multilevel analysis of predictors of multiple indicators of childhood vaccination in Nigeria "

The manuscript presents a reasonably well-articulated discussion for the developed framework to address childhood vaccination inequalities in Nigeria. Based on my reading, the applications of multilevel logistic models to produce new insights to data analysis makes the current manuscript contributing to the literature of public health. The work will be of also interest to a wide readership in the journal. Although the manuscript is worth published, its present form needs some revisions and I have the following suggestions.

Response: We appreciate your positive feedback.

1. In the Introduction, the authors omitted the brief description about multilevel models. Some details can be found at supplementary file but I think the associated discussions could be summarized into the Introduction. In so doing, the article would be more complete and clear.

Response: Thank you. We appreciate your feedback. We wish to state that the Introduction section is not where models are described and that the details of the models used in the work are already given in the Data analysis section from pages 7-9 and in the Supplementary Information section from page 7-12. Thus, we did not make any changes to the manuscript. 

2. In the Data section, the authors may provide a figure for the hieratical structure of the data, which can furnish the rationale why to use multilevel models.

Response: Thank you for this useful feedback. We now produced the suggested figure, but we prefer to put this figure in the Supplementary Information (see Fig S2 at page 7). We revised the manuscript to reflect this. See line 212-213 at page 8. 

On the other hand, for readability and the ease of understanding, the authors should at least include some model equations for the multilevel models in the section of Data analysis, rather than just assemble all the materials in the supplementary.

Response: Thank you. The authors were very careful about the need for the general audience to understand the paper. Thus, this paper hinges on both public health and statistical methodology. Our approach of sending our multilevel model equations to the Supplementary Information while describing them in the main manuscript text is strategic – (1) not to make the paper an overkill of statistical methodology which will make it very difficult for the general public health experts to understand and (2) to sustain readers interests. Fortunately, the statistical experts or modellers who are interested in these multilevel model equations can easily access them in detail in the Supplementary Information. In addition, we provided sufficient information about the multilevel models used in this paper in the main manuscript text to allow readers to follow the paper easily. As a result, we are of firm believe that is best to leave the multilevel model equations in the Supplementary Information. Thus, we did not make any changes to the manuscript. 

3. Is there any specific reason to use Bayesian multilevel models? As the ‘frequentist’ multilevel logistic regression models exist as alternative ways to analyze the data, I suggest the authors explain a bit what to motivate the Bayesian multilevel analyses and why to use them both in the Introduction and Data analysis sections.

Response: Thank you. Firstly, one of our interests is to employ Bayesian modelling approach to investigate the problem. Undoubtedly, the Bayesian modelling is one of the novel approaches in model fitting and model diagnostics. For example, in the Bayesian approach, two different sources of uncertainties (i.e., uncertainty in the parameter values and sampling uncertainty) in our estimates can be quantified and the 95% credible intervals fixed while the estimated parameters are allowed to vary unlike the frequentist (i.e., classical) approach. Also, the type of the modelling implemented in this study, especially the multiple multilevel multinomial and binomial regression models can effectively and efficiently be implemented in the Bayesian framework compared to the frequentist due to the model complexity and computational cost. These are the key reasons we opted for the Bayesian approach. However, we explore the data using cross-tabulations, and single-level logistic regression model based on frequentist approach to determine how each of the covariates relates with the outcomes. We revised the manuscript to reflect this in the Supplementary Information under the heading ‘Bayesian models’ in lines 208-219. 

Also, as presented already in the original manuscript in the ‘Supplementary Information’ under the heading ‘Prior distributions for fixed and random effects’, note that the results from the frequentist models were used to provide initial values for the Bayesian multilevel models (i.e., both the binomial and the multinomial). Thus, the results from the frequentist models were used as an input for the final Bayesian models as already captured in the Supplementary Information (see lines 193-203). 

4. Regarding the model specifications, do the multilevel models only include random intercepts? Is there any random coefficients for the predictors? From the model equation 2 and equation 3 in the supplementary file, it is not clear to me how the multilevel representation is given.

Response: Thank you. We used only random intercept multilevel models as already stated in the Supplementary Information, and we did not include random coefficient for the predictors because our goal is to examine whether the outcomes vary by the levels of hierarchy. Thus, our models did not include random slopes. This can be seen in the texts and our multilevel model equations presented in the Supplementary Information (see lines 109-111 and Equation (1) at page 8, line 120 Equation (2) at page 9, and lines 172-173, 180-181 at page 11). 

5. The authors indicated in the supplementary that non-informative priors can be alternative choice of priors for the model parameters. I was actually thinking that would it be possible to generate better analysis results as there is lack of information on reliable priors. Although I don‘t demand the authors to do so, the discussion needs to be added.

Response: Thank you. We have already discussed this in the ‘Supplementary Information’ under the heading ‘Prior distributions for fixed and random effects’ (see lines 193-203). We stated that due to lack of information on reliable priors, we fitted multilevel models via the frequentist approach and used these results as the initial values (priors) for the Bayesian models. Even though this is a reasonable approach and better than the non-informative priors approach, we were informing the readers that there is another approach where one can also use the non-informative priors, and we stated this in the same section already (see lines 193-203). Definitely, our approach of fitting the frequentist model to the data and using the parameters from these models as the priors for the Bayesian models is better than using the non-informative priors. Our approach was successfully used in a previous study [1]. We now provided the reference to support our approach in the Supplementary Information (see line 202 at page 12). 

Reference

1. Aheto JMK, Taylor BM, Keegan TJ, Diggle PJ. Modelling and forecasting spatio-temporal variation in the risk of chronic malnutrition among under-five children in Ghana. Spat Spatiotemporal Epidemiol. 2017;21:37-46. Epub 2017/03/02. doi: 10.1016/j.sste.2017.02.003. PubMed PMID: 28552186.

6. Lines 168-174: It is not clear whether the single-level logistic regression analysis was taken with Bayesian approach or not. This needs to be clarified.

Response: Thank you for the feedback. For the single level models presented, we used the frequentist approach as already stated in the original manuscript in lines 170-171 (i.e., “…fitted frequentist single level simple logistic regression models to obtain the corresponding …” but now in lines 202-205at page 7 of the revised main manuscript. Also, we already stated this in the supplementary tables. We already reported this under ‘Cross-tabulations and single-level logistic regression analysis’ in the main manuscript under ‘Data analysis’ (see lines 202-205 at page 7), and in supplementary Tables S4, S5, S7, and S9 in the Supplementary Information. 

7. Lines: 190-194: The authors did not consider interaction terms in the multinomial multilevel regression analysis. I wonder if this is due to the computational difficulty or convergence problems for the considered models? Also, does it need proportional odds assumption here? The authors may discuss these in the manuscript.

Response: Thank you. We did not consider the interaction terms for the multiple multilevel multinomial models due to model complexities and non-convergence challenges. We revised the manuscript to reflect this (see line 229 at page 8). Also, our multinomial outcomes PENTA1 and MV are nominal outcomes and not ordinal outcomes and so does not require the proportional odds assumptions.

---

## [Decision Letter · Decision Letter 1]

1 Mar 2022

PONE-D-21-28231R1Multilevel analysis of predictors of multiple indicators of childhood vaccination in NigeriaPLOS ONE

Dear Dr. Aheto,

Thank you for submitting your manuscript to PLOS ONE. After careful consideration, we feel that it has merit but does not fully meet PLOS ONE’s publication criteria as it currently stands. Therefore, we invite you to submit a revised version of the manuscript that addresses the points raised during the review process.

We look forward to receiving your revised manuscript.

Kind regards,

Tzai-Hung Wen, Ph.D.

Academic Editor

PLOS ONE

Journal Requirements:

Additional Editor Comments:

The reviews think all comments have been addressed. However, the reviewer concerns the quality and resolution of the figures. We are happy to publish the paper if the quality of the figure can be further improved.

Reviewers' comments:

Reviewer's Responses to Questions

**Comments to the Author**

1. If the authors have adequately addressed your comments raised in a previous round of review and you feel that this manuscript is now acceptable for publication, you may indicate that here to bypass the “Comments to the Author” section, enter your conflict of interest statement in the “Confidential to Editor” section, and submit your "Accept" recommendation.

Reviewer #1: All comments have been addressed

Reviewer #2: All comments have been addressed

2. Is the manuscript technically sound, and do the data support the conclusions?

Reviewer #1: Yes

Reviewer #2: Yes

3. Has the statistical analysis been performed appropriately and rigorously? 

Reviewer #1: Yes

Reviewer #2: Yes

4. Have the authors made all data underlying the findings in their manuscript fully available?

Reviewer #1: Yes

Reviewer #2: Yes

5. Is the manuscript presented in an intelligible fashion and written in standard English?

Reviewer #1: Yes

Reviewer #2: Yes

6. Review Comments to the Author

Reviewer #1: I have no more comments. However, the quality and resolutions of figures listed in this manuscript are not good enough. I suggest authors should correct them.

Reviewer #2: The revised manuscript has addressed all my concerns. I am happy to see it published in the journal.

7. PLOS authors have the option to publish the peer review history of their article (what does this mean?). If published, this will include your full peer review and any attached files.

Reviewer #1: No

Reviewer #2: No

---

## [Author Response · Author response to Decision Letter 1]

2 Mar 2022

Dear Editor,

Thank you for the opportunity to once again revise our manuscript titled “Multilevel analysis of predictors of multiple indicators of childhood vaccination in Nigeria”. We are extremely grateful to the Editors and the Reviewers for the constructive criticism of the manuscript and the positive feedback which certainly helped to improve the quality of our manuscript. 

We have duly addressed all the concerns raised by the Editors and the Reviewers to the best of our ability and wish to submit the revised version for your consideration and subsequent publication in your cherished journal, and together we can all help in addressing this serious public health challenge facing millions of children globally, especially in developing countries like Nigeria. 

Response to Editorial comments

Journal Requirements:

Response: We reviewed our reference list and can confirm that they are correct and complete. Also, we did not find any retracted paper among our references. Thus, we did not make any changes to the manuscript and the reference list. 

Additional Editor Comments:

The reviews think all comments have been addressed. However, the reviewer concerns the quality and resolution of the figures. We are happy to publish the paper if the quality of the figure can be further improved.

Response: Thank you for handling our manuscript smoothly and your decision for us to further improve our figures before our manuscript is accepted for publication. We suspect that the reviewer might not have opened the figure from the link provided at the top of each figure in the reviewer’s PDF document and appeared to have relied on what he/she saw directly on the reviewers PDF document to arrive at this concern. Please, we prepared our figures using the correct figure formats, including correct resolutions based on PLOS ONE recommended free tool (https://pacev2.apexcovantage.com/) and the quality and resolution was great on our systems. Please, may I ask the Editor to download the original figures from the link at the top of each figure in the reviewer’s PDF document for your own assessment. Always, the system that generates the reviewers document for review is responsible for the relatively low figure quality observed by the reviewer so reviewers must always download the actual figures from the link at the top of each figure which always have the quality and the required resolution from the PACE tool as recommended by PLOS ONE. 

However, we prepared new figures again using much higher resolution and used the PACE software to format the figures as recommended by PLOS ONE to improve the quality and resolution as requested. 

Per my own experience with publishing figures in PLOS ONE, and reviewing manuscripts containing figures for PLOS ONE, the quality and resolution of such figures is mostly very poor in the reviewers document even if one submitted a very high quality and resolution images/figures just as in our case now. As a result, reviewers who are not aware of this will surely raise concerns about the figures as in our case, but same figures look great when reviewers download them from the link in the reviewer’s PDF document at the top of each figure and they also look good when they are finally published in the journal once they were prepared from the PACE tool like in our case.

Reviewer #1: I have no more comments. However, the quality and resolutions of figures listed in this manuscript are not good enough. I suggest authors should correct them.

Response: Thank you. However, the reviewer might not have opened the figure from the link provided at the top of each figure in the reviewer’s PDF document and appeared to have relied on what he/she saw directly on the reviewers PDF document to arrive at this concern. Please, we prepared our figures using the correct figure formats, including correct resolutions based on PLOS ONE recommended free tool (https://pacev2.apexcovantage.com/) and the quality and resolution was great on our systems. Always, the system that generates the reviewers document for review is responsible for the relatively low figure quality observed by the reviewer so reviewers must always download the actual figures from the link at the top of each figure in the reviewers PDF document which always have the required quality and resolution from the PACE tool as recommended by PLOS ONE. 

However, we prepared new figures again using much higher resolution and used the PACE tool to format the figures as recommended by PLOS ONE to improve the quality and resolution as suggested. 

Also, note that we now combined Figures 1 and 2 to form Figure 1 because they are both coming from the same model, and we thought is better to present them as one figure. Accordingly, we revised the manuscript to correct the figure numbers and added few text (see lines 270, 273, and 274 at page 10, line 285-286, and 294 at page 11, and lines 302-309 at page 12, lines 332 and 344 at page 13). 

Reviewer #2: The revised manuscript has addressed all my concerns. I am happy to see it published in the journal.

Response: Thank you.

---

## [Editor Report · Decision Letter 2]

28 Mar 2022

PONE-D-21-28231R2Multilevel analysis of predictors of multiple indicators of childhood vaccination in NigeriaPLOS ONE

Dear Dr. Aheto,

Thank you for submitting your manuscript to PLOS ONE. After careful consideration, we feel that it has merit but does not fully meet PLOS ONE’s publication criteria as it currently stands. Therefore, we invite you to submit a revised version of the manuscript that addresses the points raised during the review process.

We look forward to receiving your revised manuscript.

Kind regards,

Tzai-Hung Wen, Ph.D.

Academic Editor

PLOS ONE

Journal Requirements:

Additional Editor Comments:

The quality and resolutions of figures listed in this manuscript are not good enough. I suggest authors should correct them.

---

## [Author Response · Author response to Decision Letter 2]

28 Apr 2022

Dear Editor,

Thank you for the opportunity to once again revise our manuscript titled “Multilevel analysis of predictors of multiple indicators of childhood vaccination in Nigeria”. We are extremely grateful to the Editors and the Reviewers for the constructive criticism of the manuscript and the positive feedback which certainly helped to improve the quality of our manuscript. 

We have duly addressed all the concerns raised by the Editors and the Reviewers to the best of our ability and wish to submit the revised version for your consideration and subsequent publication in your cherished journal, and together we can all help in addressing this serious public health challenge facing millions of children globally, especially in developing countries like Nigeria. 

Response to Editorial comments

Journal Requirements:

Response: We reviewed our reference list and can confirm that they are correct and complete. Also, we did not find any retracted paper among our references. Thus, we did not make any changes to the manuscript and the reference list. 

Additional Editor Comments:

The quality and resolutions of figures listed in this manuscript are not good enough. I suggest authors should correct them.

Response: Thank you for handling our manuscript smoothly and your decision for us to further improve our figures before our manuscript is accepted for publication. 

We have prepared our figures following the recommended guidelines from PLOS ONE, including correct resolutions based on the recommended free tool (https://pacev2.apexcovantage.com/) and the quality and resolution was great on our systems (see original figures in the PLOS ONE submission system).

We believe our figures were produced with the required quality and resolution. However, we suspect that the Reviewers might not have accessed the original figures provided via the links at the top of each figure in the PDF proof of submission which was reviewed by the reviewers. This might have led them to persistently request that we produce new figures with high quality and resolution. 

In our first revision, we responded to item 7 from reviewer #1 and the query was “Figures: The quality and resolution of figures are very bad. It’s hard to know the results.” Clearly, for reviewer #1 to say “… It’s hard to know the results”, the reviewer was only relying on the figures as appeared in the PDF Proof of submission (images in the PDF proof) rather than accessing the original figures embedded in the proof links above each figure. Nonetheless, we improved the quality and resolution of our figures in our revised submission dated on 3rd March 2022, but the comments on the quality of figures came up in subsequent decision letter. Also, we have previously written to the Editor on 2nd March 2022 but we are yet to receive any response except a new request to improve our figures. 

As presented below, the reviewers seem to be basing their quality assessment of our figures on Fig 2 (i.e., the direct images shown in the pdf proof) instead of assessing Fig 1 (i.e., the original images embedded in the PDF proof links at the top right of each figure).

Fig 1 Screenshot of Figure 1 from the link on the top right of Figure 1 in the PDF Proof of submission (see the Response to Reviewers letter submitted on word document via the submission system for the figures).

Fig 2. Screenshot of the image shown in Figure 1 directly from the PDF Proof of submission (see the Response to Reviewers letter submitted on word document via the submission system for the figures).

Per the above evidence, our figures have good quality and resolution, and we believe strongly that our figures met PLOS ONE requirements to be accepted for publication. Thus, we did not make any changes to the figures after the second revision we have done to improve it further. 

Note: we have now included acknowledgement section to our revised manuscript (see lines 456-457 at page 17).

---

## [Editor Report · Decision Letter 3]

16 May 2022

Multilevel analysis of predictors of multiple indicators of childhood vaccination in Nigeria

PONE-D-21-28231R3

Dear Dr. Aheto,

We’re pleased to inform you that your manuscript has been judged scientifically suitable for publication and will be formally accepted for publication once it meets all outstanding technical requirements.

Kind regards,

Tzai-Hung Wen, Ph.D.

Academic Editor

PLOS ONE
---

## [Editor Report · Acceptance letter]

17 May 2022

PONE-D-21-28231R3 

Multilevel analysis of predictors of multiple indicators of childhood vaccination in Nigeria 

Dear Dr. Aheto:

I'm pleased to inform you that your manuscript has been deemed suitable for publication in PLOS ONE. Congratulations! Your manuscript is now with our production department. 

Kind regards, 

on behalf of

Dr. Tzai-Hung Wen 

Academic Editor

PLOS ONE